# Diversity, Distribution, and Biogeography of Freshwater Fishes in Guangxi, China

**DOI:** 10.3390/ani12131626

**Published:** 2022-06-24

**Authors:** Jiayang He, Zhiqiang Wu, Liangliang Huang, Minhui Gao, Hao Liu, Yangyan Sun, Saeed Rad, Lina Du

**Affiliations:** 1College of Environmental Science and Engineering, Guilin University of Technology, Guilin 541004, China; hjy2022wsry@163.com (J.H.); wuzhiqiang@glut.edu.cn (Z.W.); liuhaoixx@163.com (H.L.); syy251705@163.com (Y.S.); saeedrad1979@gmail.com (S.R.); 2Innovation Center for Water Pollution Control and Water Safety Guarantee in Karst Areas, Guilin University of Technology, Guilin 541004, China; 3College of Life Science and Technology, Guangxi University, Nanning 350000, China; gmhxxzj@126.com; 4College of Life Sciences, Guangxi Normal University, Guilin 541004, China; dulina@mailbox.gxnu.edu.cn

**Keywords:** freshwater fish, beta diversity, taxonomic diversity, cavefish, biogeography

## Abstract

**Simple Summary:**

The Guangxi Zhuang Autonomous Region has one of the most abundant aquatic biodiversity in China, and it is a hotspot of global biodiversity research. In the present study, we explored the diversity, distribution, and biogeography of freshwater fishes in Guangxi. Our results showed that 380 species of freshwater fishes were recorded in Guangxi; the species diversity from northwest to southeast gradually decreased for most Sub−basins; the spatial turnover component was the main contributor to beta diversity; the freshwater fish system belonged to the South China division in the Southeast Asiatic subregion of the Oriental region.

**Abstract:**

The Guangxi Zhuang Autonomous Region has the largest number of cavefish species in the world and is a global biodiversity hotspot. In this study, a species list of freshwater fishes in 12 Sub−basins of Guangxi was compiled systematically. Moreover, the species composition and distribution of the diversity were analyzed via the *G-F* index, taxonomic diversity index, and beta diversity index. Results showed that 380 species of freshwater fishes were recorded in this region, which belonged to 158 genera in 43 families and 17 orders in 2 phyla, in which 128 species of endemic fishes and 83 species of cavefish accounted for 33.68% and 21.84%, respectively. The species diversity from northwest to southeast gradually decreased for most Sub−basins. The *G-F* index has generally risen in recent years. The taxonomic diversity index showed that the freshwater fish taxonomic composition in Guangxi is uneven. The spatial turnover component was the main contributor to beta diversity. A cluster analysis showed that the 12 Sub−basins in the study area could be divided into four groups, and the phylogenetic relationships of freshwater fishes in Guangxi generally reflect the connections between water systems and geological history. The freshwater fish system in Guangxi, which belonged to the South China division in the Southeast Asiatic subregion of the Oriental region, originated in the early Tertiary period. The results will provide the information needed for freshwater fish resource protection in Guangxi and a reference for promoting the normalization of fish diversity conservation in the Pearl River Basin and other basins.

## 1. Introduction

Freshwater fishes are the most affected by human activities worldwide [1,2], yet freshwater fish diversity conservation has not received the same attention as other vertebrates [3]. Species diversity is central to biodiversity conservation [4] and, therefore, accurately describing, measuring, and assessing species diversity are long-standing concerns in ecology and conservation biology [5,6]. Species diversity indices, including the Shannon index, Margalef abundance index, and Pielou evenness index, are extremely sensitive and can be affected by different sampling methods [7]. In contrast, the mean taxonomic difference index and taxonomic difference variation index [8], the *G-F* index (generic family diversity index) [9], and the beta diversity index [10] can calculate biodiversity quickly, based on binary data of species presence or absence, without relying on sampling methods. Two processes—turnover and nestedness—are comprised of beta diversity [11]. Beta diversity measures variation in species composition among habitats, providing an overview of the similarity degree between communities [11]. The spatial variations in species compositions allow the testing of hypotheses about the processes that generate and maintain biodiversity in ecosystems, which is extremely important information to be considered by protected-area planners to establish regions of great interest for conservation [12]. Therefore, these indices are widely used in different aquatic ecosystems [13,14].

The Guangxi Zhuang Autonomous Region (hereinafter referred to as Guangxi) is located at the southern border of China, and its rivers belong to four major basins (the Pearl River basin, the Yangtze River basin, the Red River basin, and the rivers flowing into the Beibu Gulf) spanning three climatic zones (the northern tropics, the southern subtropics, and the central subtropics) with well-developed karst topography [15]. Guangxi, as one of the global biodiversity hotspots, is one of the richest areas in aquatic biodiversity and an important freshwater fishery production base in China [16].

From the first report on the new genus and species (*Sinohomaloptera kwangsiensis*) of Guangxi in the 1930s to the publication of *Freshwater Fishes of Guangxi* (second edition) in 2006, a total of 290 species and subspecies of freshwater fishes in Guangxi were recorded [15]. Later, works such as *Cave Fishes of Guangxi, China* [17], *Fishes of the Pearl River* [18], and *Investigation and Research on main River Fish Resources in Guangxi of the Pearl River Basin* [19] were successively published.

In recent years, fish biodiversity, community structure, and fish integrity in Guangxi have emerged [20,21,22,23], and new species and non-native species have also been reported [16,24]. Moreover, the taxonomic statuses of some species have been corrected [25,26]. The diversity of freshwater fishes in Guangxi has changed due to increasing problems, such as soil erosion, massive dam construction, water pollution, and alien species invasion [23]. Therefore, based on previous literature and our data, regarding the latest taxonomy and molecular systematic research on fish, the main aims of the present study were: (1) to determine the freshwater fish species composition and distribution in Guangxi; (2) to clarify what contributes the most to beta diversity—replacement or nestedness; (3) to explore the formation of spatial distribution patterns and fish faunal compositions of freshwater fish in Guangxi.

## 2. Materials and Methods

### 2.1. Study Area

The Guangxi Zhuang Autonomous Region, located in the southern frontier of China, has nearly 1000 rivers and abundant wild fish resources. It is a key area of tropical and subtropical biodiversity in China. The whole terrain of Guangxi tilts from northwest to southeast, and rivers flow along the slope of the terrain. The river source water level is high with many gorges, rapids, and deep pools. The lower reaches of the river become gradually wider with a gentler water flow. The karst landform is greatly distributed in the territory, with surface water leakage, and a well-developed underground river system. Its average precipitation and temperature ranges are 1100–1890 mm and 17.1–21.9 °C, respectively (Table 1). The rivers are divided into four major basins: the Pearl River Basin, the Yangtze River Basin, the Red River Basin, and the rivers flowing into the Beibu Gulf (Figure 1).

### 2.2. Data Sources

Based on *Freshwater Fishes of Guangxi* [15], the species composition of freshwater fishes in Guangxi was counted and revised according to the latest taxonomic and molecular systematic research results, and the list of freshwater fishes in Guangxi was formed. The Latin names refer to the latest molecular systematic research [27]. The validity and classification statuses of all species were confirmed in the Catalog of Fishes database [27]. Species distribution data were arranged in the Sub−basin presence/absence form. The threatened species classification included extremely critical (CR), endangered (EN), and nearly critical (VU) [28]. Statistical results are shown in Appendix A.

### 2.3. Data Processing

The freshwater basins in Guangxi were divided into 12 Sub−basins by combining topography, geomorphology, and administrative division [15]. ArcMap 10.2 software was used to divide the basins into different zones (Figure 1). To analyze the significant differences in fish species numbers between years and subbasins, the paired-sample T test was used for statistical analysis by SPSS 21.0. The average taxonomic difference index (Δ^+^), and taxonomic difference variation index (*Λ*^+^) were used to calculate freshwater fish diversity in Guangxi in 2006 and 2021 [8].

The Genus Family index (*G-F* index) derived from the Shannon diversity index, has been successfully used to assess bird and mammal biodiversity [9]. The standardized *G-F* index was calculated based on biodiversity values at the genus level (*G* index) and family level (Findex), as follows:

*G* index:$$D_{G}=-\sum_{j=1}^{p}D_{Gi}\;=-\sum_{j=1}^{p}\;\frac{S_{j}}{S}\;ln\frac{S_{j}}{S}$$
where *S*_j_ is the number of species in the genus *j*, *S* is the total number of species in the class, and *p* is the number of genera in the class.

*F* index:$$D_{F}=-\sum_{k=1}^{m}\;\sum_{i=1}^{n}\;\frac{S_{ki}}{S_{k}}\;ln\;\frac{S_{ki}}{S_{k}}$$
where *n* is the number of genera in the family *k*, m is the total number of families in the class, *S_ki_* is the number of species in genus *i*, and S_k_ is the total number of species in family *k*.

*G-F* index:$$D_{G-F}=1-\;\frac{D_{G}}{D_{F}}$$

In this study, species checklist data were used to calculate these indices to assess the species diversity in each survey area. The *G* index reflects the diversity at the genus level, and the *F* index reflects the diversity both within and among families. The *G-F* index usually ranges between 0 and 1; if there is only one species in the survey area or a few belong to different families, then the *G-F* index is defined as 0 [9].

Taxonomic distinctness was proposed by Warwick and Clarke [8], in which the average taxonomic distinctness (Δ^+^) and the variation in taxonomic distinctness (*Λ*^+^) are used to evaluate the distance between the taxonomy of the species and the distance of the hierarchical taxonomic tree, as follows:

Average taxonomic distinctness:
$$\Delta^{+}=(\sum \;\sum_{i<j}\;w_{ij})/\left[S\;\left(S-1\right)/2\right]$$

Variation in taxonomic distinctness:$$\Lambda^{+}={\left(\sum \;\sum_{i<j}\;w_{ij}-\Delta^{+}\right)}^{2}/\left[S\;\left(S-1\right)/2\right]$$
where *w_ij_* is the distinctness mass given to the path length linking species *i* and *j* in the hierarchical classification, and *S* is the total number of fish species in the survey.

In this work, the classification of fish was divided into five levels: class, order, family, genus, and species. The weights of the path lengths of the species belonging to the same phylum but not the same class; the same class but not the same order; the same family but not the same genus; and the same genus but not the same species were 83.333, 66.667, 50.000, 33.333, and 16.667, respectively [8]. Furthermore, the values of Δ^+^ and *Λ*^+^ were tested for departure from the expectation according to randomization tests. A randomization test with 10,000 random selections was used to detect the expected values of Δ^+^ and *Λ*^+^ derived from the species pool (master list), enabling us to test the significance of departure between the observed and expected values of the two indices. Such plots are described as confidence funnel plots, where degraded sites are assumed to fall below the lower 95% confidence limits, while reference sites should be located within the 95% confidence limits [29]. The calculations of Δ^+^ and *Λ*^+^ and the randomization test were computed using the TAXDTEST procedure in PRIMER 5.0 [29].

Beta diversity is represented by the differences in species compositions between different communities, determined by species turnover (species replacement) and nestedness (richness difference) [30]. To quantify the effects of the two processes, Baselga [31] systematically proposed the beta diversity decomposition method (BAS frameworks) based on the Sørensen index (β_sor_), which was decomposed into species spatial turnover components (β_sim_) and nestedness components (β_sne_), as well as the Jaccard index (β_jac_), which was decomposed into species spatial turnover components (β_jtu_) and nestedness components (β_jne_).

Sørensen index:$$\beta_{\mathrm{sor}}=\frac{b+c}{2a+b+c};\;\beta_{\mathrm{sim}}=\frac{\min\left(b,c\right)}{a+\min\left(b,c\right)};\;\beta_{\mathrm{sne}}=\frac{\left|b-c\right|}{2a+b+c}\times \frac{a}{a+\min\left(b,c\right)}$$

Jaccard index: $$\beta_{\mathrm{jac}}=\frac{b+c}{a+b+c};\;\beta_{\mathrm{jtu}}=\frac{2\min\left(b,c\right)}{\;a+2\min\left(b,c\right)};\;\beta_{\mathrm{jne}}=\frac{\left|b-c\right|}{a+b+c}\times \frac{a}{a+2\min\left(b,c\right)}$$

A dataset covering all species recorded in each Sub−basin was then constructed, and similarity analyses were carried out based on a logarithmic transformation for the number of each species in each Sub−basin. Pairwise similarities among Sub−basins were then computed to create a similarity coefficient matrix. The hierarchical cluster and the furthest-neighbor method with the Bray–Curtis similarity were then used for the cluster analysis based on the matrix. The above analyses were calculated using PRIMER 5.0.

We performed Mantel tests and partial Mantel tests [32] with 9999 permutations to assess the correlations (Spearman’s method) between eight pairwise similarity matrices and the matrices of geographical drivers (watershed area, river length, average precipitation, annual average runoff, average altitude, average gradient, and average temperature among streams; Table 1) to explore the potential mechanisms that explained the beta-diversity patterns. The partial Mantel tests were used to remove the effect of covariation because intercorrelations between matrices of differences in the watershed area, river length, average precipitation, annual average runoff, average altitude, average gradient, and the average temperature were detected (*p* < 0.05). The above analyses were performed in R 4.1.0 using the BETAPART package [31] and VEGAN [33].

## 3. Results

### 3.1. Faunal Composition

A total of 380 freshwater fish species, belonging to 17 orders, 43 families, and 158 genera, have been recorded in the freshwater and estuarine areas of Guangxi. Among these, 128 endemic species, 83 species of cavefish, 49 threatened species, and 18 alien species were recorded, accounting for 33.68%, 21.84%, 12.89%, and 4.74% of the total freshwater fishes in Guangxi, respectively. The updated list added 94 newly recorded species, including 13 newly recorded alien species and 81 newly undescribed species. The natural distribution of native freshwater fish (estuarine fish and migratory fish were excluded) in Guangxi is 342 species, 129 genera, 19 families, and 5 orders (Appendix A). Cypriniformes constitute the main body of freshwater fish fauna in Guangxi, with 281 species belonging to 103 genera and 3 families, accounting for 81.92% of the total native freshwater fishes in the study area. Siluriformes is the second, with 29 species belonging to 13 genera and 6 families (8.45%). Perciformes has 27 species of 12 genera and 7 families, accounting for 7.87% of the total. There are 2 families, 2 genera, and 3 species of Synbranchiformes. The order Beloniformes has 1 family, 1 genus, and 2 species. Ostariophysi, composed of Cypriniformes and Siluriformes, accounts for 90.38% in total and 10.62% in other orders. Compared to 2006, the number of native freshwater fish species in Guangxi increased by 32.56% in 2021, among which, HR, LGR, HSR, LR, XZR, YR, and ZR, compared with other Sub−basin growth rates, increased significantly (*p* < 0.05, Figure 2).

### 3.2. Fish Distribution

The number of freshwater fish species in each Sub−basin decreased as the latitude decreased, generally showing a declining trend from northwest to southeast (Figure 3). The results of the paired-sample T test indicated that the species of OR and NLR were slightly lower, and the northern Sub−basins (HSR, LR, and GLR) had markedly more fish species than OR and NLR (*p* < 0.05). The LGR had the highest number of species (185), genera (96), and families (17), while BDR had the lowest (18 species, 17 genera, and 5 families).

The top 8 genera ranked by the number of fish species are *Sinocyclocheilus*, *Troglonectes*, *Triplophysa*, *Pseudobagrus*, *Microphysogobio*, *Acrossocheilus*, *Onychostoma*, and *Parabotia*. Frequency of distribution, *Schistura fasciolatus*, *Opsariichthys bidens,* and *Hemibarbus maculatus* occurred in all 12 Sub−basins, followed by *Traccatichthys pulcher*, *Hemiculter leucisculus*, *Pseudohemiculter dispar*, *Hemibarbus labeo*, *Pseudorasbora parva*, *Squalidus argentatus*, *Acheilognathus tonkinensis*, *Puntius semifasciolatus*, *Onychostoma gerlachi*, *Cyprinus rubrofuscus*, *Carassius auratus*, *Silurus asotus*, *Pseudobagrus crassilabris*, *Hemibagrus guttatus*, *Mastacembelus armatus*, *Rhinogobius giurinus*, *Channa maculata*, and *Channa asiatica* occurring in 11 Sub−basins. The fewest are *Paranemachilus genilepis*, *Sinocyclocheilus guilinensis*, and *Bagarius yarrelli*, with 143 other species, occurring in only one Sub−basin.

There are 110 genera of fishes in Guangxi, distributed in two or more water systems at the same time; the remaining 19 genera of fishes are distributed only in a single water system. The genera distributed only in HSR in Guangxi are *Micronemacheilus*, *Yunnanilus*, *Lanlabeo*, *Pseudogyrinocheilus*, *Hongshuia*, *Discocheilus,* and *Paraprotomyzon*; the genera distributed only in LGR are *Stenorynchoacrum* and *Yaoshania*; *Tanichthys* in XR only; *Atrilinea* in LR only; *Zuojiangia*, *Sinigarra,* and *Prolixicheilus* in ZR only; *Parazacco*, *Pogobrama*, and *Anabacco* in OR only, *Pogobrama*, *Anabas*, and *Mystus* in OR only; and lastly, *Bagarius* in BDR only.

In addition, there are 83 species of cavefish in Guangxi belonging to 14 genera, 4 families, and 2 orders. The *Sinocyclocheilus* is the largest group in Cyprinidae, with 34 species, followed by 14 species of *Troglonectes* in Nemacheilidae, 11 species of *Triplophysa*, 5 species of *Heminoemacheilus*, 5 species of *Oreonectes*, 3 species of *Paranemachilus*, 3 species of *Protocobitis*, 2 species of *Schistura*, and 1 species of *Traccatichthys*, *Micronemacheilus*, *Yunnanilus*, *Bibarba*, *Parasinilabeo,* and *Xiurenbagrus*. These cavefish are only distributed in 6 Sub−basins, of which, 39 species in HSR account for 50.65% of the total species; 25 species in LR account for 32.47% of the total species; 7 species in LGR and 7 species in YR account for 9.09% of the total species. Five species in HR account for 6.49% of the total species and four species in ZR account for 5.19% of the total species.

### 3.3. Biodiversity at Three Taxa Levels

The *F* index, *G* index, and *G*-*F* index in each Sub−basin were calculated (Figure 4). Compared with 2006, the *F* indices of nine Sub−basins, in general, showed upward trends, except for YYJ, NLJ, and OR. Moreover, LR had the highest value of 12.14 and BDR had the lowest value of 4.09. The *G* indices of nine Sub−basins, except for YYR, HSR, and NLR, showed upward trends, with the highest value of LR being 4.36 and the lowest value of BDR being 2.81. The *G*-*F* indices of 11 Sub−basins, except NLR, increased from 2006 to 2021, and the *G*-*F* index decreased from the northwest (LR, HSR, and LGR were 0.64) to all directions. In general, the *F* index (14.74) and *G*-*F* index (0.70) of freshwater fishes in 2021 were higher than those in 2006 (13.37 and 0.67), while the *G* index (4.43) in 2021 was lower than that in 2006 (4.47).

### 3.4. Taxonomic Diversity Index

The average classification difference index (Δ^+^) for 2006 and 2021 ranged from 40.2 to 48.0 and 40.6–48.3, with theoretical mean values of 42.8 and 43.2, respectively (Table 2). The classification difference variation index (*Λ*^+^) for 2006 and 2021 ranged from 333.1 to 484.2 and 381.2 to 480.9 and the theoretical mean values were 431.3 and 434.5, respectively (Figure 5). The Δ^+^ of XZR was the largest in 2021, followed by OR, HR, and BDR, indicating that fish in these Sub−basins had a long genetic relationship and high taxonomic diversity. The Δ^+^ of NLR was the smallest, followed by YR and HSR, indicating that fish were closely related and had low taxonomic diversity in these basins. XR had the largest *Λ*^+^, followed by YYR, HR, OR, and NLR, indicating that fish in these Sub−basins had the most uneven classification orders. BDR had the smallest *Λ*^+^, followed by HSR, LGR, and XZR, indicating that fish had the most uniform classification orders in these basins.

### 3.5. Beta Diversity Patterns

The fish composition similarities in 12 Sub−basins of Guangxi had mean values of 0.79 and 0.88, based on Sørensen and Jaccard indexes, orderly (SD ± 0.05 and SD ± 0.05, respectively; Table 3). The spatial turnovers and replacement components (β_sim_ and β_jtu_, 0.60 ± 0.06, and 0.75 ± 0.05) were greater than the nestedness and richness difference components (β_sne_ and β_jne_, 0.19 ± 0.03 and 0.13 ± 0.03). BDR and HR had high β_sor_ and β_jac_ (0.83 ± 0.03 and 0.91 ± 0.02; 0.55 ± 0.11 and 0.70 ± 0.08). Moreover, high spatial turnover and replacement components (0.40 ± 0.10 and 0.57 ± 0.10) in XZR and nestedness and richness difference components (0.28 ± 0.18 and 0.31 ± 0.18) in LGR (Table 3) were observed.

The results of the cluster analysis of the species similarity in 12 Sub−basins based on the Jaccard similarity coefficient showed that it could be divided into four groups when the similarity coefficient was 50% (Figure 6). Group 1 was composed of the BDR; *Traccatichthys taeniatus*, *Garra imberba,* and *Bagarius yarrelli* were only distributed in this Sub−basin of Guangxi. Group 2 was HJ and there was no endemic genus in this Sub−basin. Group 3 included the OR, NLR, XR, YYR, YR, and ZR, among which, the OR and NLR of the southern rivers into the Beibu Gulf were clustered in one. ZR and YR were clustered in one; *Paranemachilus* was distributed only in ZR. YR, XR, and YYR of the main streams of the Xijiang River basin were clustered in one. Lastly, Group 4 included HSR, LR, LGR, and XZR; *Acrossocheilus*, *Onychostoma*, *Microphysogobio*, *Sinocyclocheilus*, *Parasinilabeo*, *Heminoemacheilus*, *Oreonectes*, *Troglonectes,* and *Triplophysa* were the main reasons for the difference in fish compositions between this group and other groups.

We found that the correlation between the Sørensen index and the Jaccard index, as well as the difference in the length, area, average gradient, average temperature, and annual average runoff, were not significant (*p* > 0.05) in 12 Sub−basins. The correlation between β_sor_ (β_jac_) and the difference in the average altitude was significant (*p* < 0.05). The correlation between β_sim_ (β_jac_, β_sne_, and β_jne_) and the difference in the average precipitation was also significant (*p* < 0.05) (Figure 7).

## 4. Discussion

### 4.1. Change of Freshwater Fish Species Composition

The updated list of freshwater fishes in Guangxi contains 380 fish species, accounting for 23.53% of the number of freshwater fish species in China (1615 species, quoted from the website: www.fishbase.org (accessed on 11 May 2022)), which is the second-highest among all provinces in China [34]. There are 83 species of cavefish in Guangxi, ranking first in China and even worldwide [17]. The updated list added 81 newly undescribed species; because the HSR, LR, and LGR water systems are located in areas with highly developed karst landscapes on the Yunnan–Guizhou Plateau, the surface and groundwater habitat diversities are extremely high, providing suitable environmental conditions for strong differentiation between surface-dwelling and burrowing fishes [35,36]. For example, *Sinocyclocheilus*, *Hongshuia*, *Oreonectes*, *Troglonectes,* and *Triplophysa* are the most strongly differentiated genera of fish species in the Yunnan–Guizhou Plateau.

Moreover, the updated list added 13 newly recorded exotic species, with the acceleration of global economic integration, aquaculture, fishery ornamental, and water ecological compensation caused by the introduction (more frequently) between countries or river systems, making freshwater fish invasion increasingly serious [37,38]. Studies have shown that invasive fish are distributed in all major rivers in Guangxi, and some invasive fish have successfully established natural populations, becoming dominant species in the invasive river segment [39]. The invasion of exotic fishes can destroy the ecological characteristics of the recipient sites, compete for the food of indigenous fishes, and compress the survival spaces of indigenous fishes [40]. Therefore, the regulation of exotic species merits concern.

### 4.2. Spatial Patterns of Fish Diversity

The karst habitats within the territory of Guangxi are diverse and complex. With more underground rivers and lakes, karstic waters are endemic, providing special external conditions for fish. Poor light is a good example of such conditions, making cavefish specious-adapted. About 85% of cavefish, such as HSR, LR, and LGR, are distributed in northwest Guangxi, and genetic exchanges between them are practically difficult [41]. Therefore, there are many endemic genera and endemic species in the northwestern parts of the region. Furthermore, modern physical geographic conditions, such as topography, altitude, and moisture limitation, affect the reproductive differentiation of freshwater fishes. Among the effects of watershed characteristics on fish diversity, the influence of altitude is extremely strong [42]. Altitude reflects changes in temperature, precipitation, primary productivity, food availability, and human activity [43]. The altitude of each Sub−basin in Guangxi decreased from northwest to southeast, and the diversity of freshwater fish in Guangxi decreased monotonously from northwest to southeast, the spatial distribution patterns were similar to those in the middle and lower reaches of the Yangtze River [44].

The *G-F* index of freshwater fishes in Guangxi (0.70) was higher than that in 2006 (0.67), indicating that the diversity of freshwater fishes in Guangxi at the family and family genus levels is becoming greater over time. Some studies have suggested that the *G-F* index is related to the number of species [45], which is consistent with the results of this study. The *D*_G_ index (4.43) was lower than that of 2006 (4.47), and the decrease was related to the increasing number of new species under the original genus of freshwater fishes in Guangxi in recent years, such as *Sinocyclocheilus* (17 new species), *Triplophysa* (8 new species), and *Troglonectes* (12 new species). While the *D*_F_ index (14.74) was higher than that of 2006 (13.37), the rise was related to the addition and establishment of a single new genus of freshwater fishes, such as *Tanichthys* [46], *Lanlabeo* [47], *Zuojiangia* [48], *Sinigarra* [49], *Stenorynchoacrum* [50], etc. In addition, LJ, HSR, and LGJ had the highest *G-F* indices (0.64), which are consistent with the high number of taxonomic orders and species at the family and genus levels in these three Sub−basins.

The Δ^+^ and *Λ*^+^ of freshwater fishes in 2021 (43.2 and 434.5) were higher than that in 2006 (42.8 and 431.5); the upward trend of Δ^+^ indicates that fish species have become more distantly related to each other and taxonomic diversity is increasing. Moreover, the increase of *Λ*^+^ underpinned that fish species in classification element distributions are more uneven. These were related to the establishment of new genera and newly recorded exotic fishes of the freshwater fishes in Guangxi in recent years. It is generally accepted that in communities of equal species compositions, the biodiversity of communities with species belonging to multiple genera is higher than that of belonging to one genus [51]. In addition, Guangxi is the transit station and main market for ornamental fish imports in China [40]. A large number of exotic fishes have been introduced, bred, discarded, or escaped [40]. Its warm climate and dense river network not only promote the breeding of exotic fish, but also facilitate the survival, reproduction, and diffusion of exotic fishes after escape [23,41]. Moreover, Guangxi is rich in hydropower resources [15]; for example, eleven cascade power stations had been planned for Hongshui River [23], which forms the river section of the reservoir area, resulting in the rise of water levels, a decrease of velocity, an increase of nutrients, improvement of primary productivity of the water body, and provision of abundant bait sources for the colonization and diffusion of exotic fishes [37,38].

Meanwhile, OR and XZR showed higher Δ^+^ values in 2021, which means the classification levels of species in these Sub−basins are relatively high. The Δ^+^ values of YR and NLJ were below the average of the funnel in 2021, indicating that the fish population structure in these sub-watersheds is relatively simple, which may be due to the fact that the homogeneous habitat (no karst landform was formed) of these Sub−basins is unable to provide conditions for freshwater fish differentiation, which greatly affects the fish species composition and taxonomic diversity [14].

### 4.3. Biogeographic Analysis of Fish

The 12 Sub−basins were divided into four groups based on Jaccard’s fish similarity coefficient, including: BDR of the Red River system; HJ without endemic genera; OR, NLR, XR, YYR, YR, and ZR; and lastly HSR, LR, LGR, and XZR. The occurrence of these groupings may be associated with geological events, such as plate tectonic movements [35]. These movements produce geographic isolation, which can affect the evolution of freshwater fishes [52].

The main section of the Xijiang River was developed by the Yanshan movement in the late Mesozoic when it was not connected with the water system of the Guangxi Basin. By the Tertiary Himalaya movement, the Yunnan–Guizhou Plateau rose and the Xijiang Valley (now YYR) sank [35]. These, together with the uplift of the Shiwan Mountain, caused the current river system on the south bank of the Xijiang River to be separated from the river flowing into the Beibu Gulf [53]. Compared to other Sub−basins in Guangxi, the rivers flowing into the Beibu Gulf gave birth to some unique fish, such as *Parabotia parva*, *Parazacco fasciatus*, *Pogobrama barbatula*, *Xenocyprioides parvulus*, *Squalidus atromaculatus*, *Anabas testudineus,* and *Channa nox*. Therefore, the rivers that flow into the Beibu Gulf (OR and NLR) are clustered in one. With the continuation of neotectonic movements, the Guangxi basin first rose extensively; however, it was still influenced by the rising of the Yunnan–Guizhou plateau, so the basin in the northwestern part of Guangxi rose more. As a result, the horizontal plane from northwest to southeast slowly inclined, and gradually formed a continuous river system [35]. Hence, the YR and ZR gathered in one group, and the XR and YYR as the main streams of the Xijiang River basin gathered in one group. At that time, the rising Duyang Mountain and Daming Mountain formed the watershed of HSR and YR, and the Sub−basins of northwest Guangxi (HSR, LR, and LGR) could not flow into the Xijiang River (temporarily) [35]. Whereafter, under the control and influence of the fourth act of the Himalayan Movement, the Guangxi northwest Sub−basins entered the stage of large-scale karst area formation and formed a new unified underground river system [36]. Meanwhile, the surface and underground rivers in the Sub−basins, where each karst area was located, gradually attacked and connected, which shaped a unified and complex hydrogeological unit [36]. At this stage, the Cyprinidae and Nemacheilidae fishes diverged rapidly; *Sinocyclocheilus*, *Heminoemacheilus*, *Oreonectes*, *Troglonectes,* and *Triplophysa* were adaptively evolving, leading to the main genus of fish composition differences between these and other groups. Because of the low base and deep depression of the Xijiang Valley, the HSR, LR, and LGR were attracted toward converging into the Xijiang River [35]. Overall, these evolutionary changes caused the distribution patterns and community compositions of freshwater fishes in Guangxi in the present era.

The freshwater fish system of China belongs to the Oriental region and Holarctic region, while Guangxi entirely belongs to the Oriental region [54,55]. Chen [54] considered the line between the Himalayas and the Qinling Mountains as the real boundary between the Palearctic and the Oriental regions in East Asia, from the perspective of the historical development of fauna. At the same time, according to the natural distribution area of East Asian fish groups, the Oriental region of East Asia is divided into the South Asiatic subregion and the Southeast Asiatic subregion based on the eastern margin of the Yunnan plateau. The other East Asiatic subregions of the eastern edge of the Yunnan Plateau, including most of the Pearl River Basin, belong to the Southeast Asiatic subregion. In view of the vast area of the Southeast Asiatic subregion, the variability of fish fauna, and physical geography, Chen also divided this subregion into the East China division in the north (including the Yangtze River and Huaihe River) and the South China subregion in the south (including parts of the Pearl River system, Zhejiang–Fujian River systems, Hainan Island River system, and Yuanjiang River system), with Miaoling Mountains, Nanling Mountains, and Wuyi Mountains as the boundary [54]. Xiangjiang River and Zishui River in Guangxi belong to the Dongting Lake water system in the Yangtze River basin and are located in the Nanling mountains. Their fish system accounts for 27.78% of the total number of native freshwater fishes in Guangxi, but their fish system lacks the distribution of endemic families (Polyodontidae, Salmonidae, and Catostomidae), the endemic genera (*Coreius*, *Ancherythroculter*, and *Pseudobrama*), and the endemic species (*Acrossocheilus monticolus*, *Ancherythroculter wangi*, *Coreius guichenoti*, *Gobiobotia filifer*, *Percocypris pingi*, *Procypris rabaudi*, *Pseudogyrinocheilus prochilus*, *Rhinogobio hunanensis*, *Schizothorax kozlovi*, *Xenophysogobio boulengeri*, *Leptobotia elongate*, *Homatula potanini*, *Jinshaia abbreviata*, *Liobagrus marginatu*) of the Yangtze River [54]. Meanwhile, endemic species of Yangtze fishes widely distributed in the Dongting Lake water system, such as *Chanodichthys oxycephaloides*, *Microphysogobio tungtingensis*, *Rectoris luxiensis*, *Sinilabeo tungting*, *Leptobotia tchangi*, *Leptobotia tientai Leptobotia tientaiensis hansuiensis*, *Parabotia banarescui*, *Lepturichthys fimbriata,* and *Sinogastromyzon hsiashiensis*, do not have distribution in the Xiangjiang River and Zishui River in Guangxi [56]. This indicates that the Nanling Mountains, which straddle the Xiangjiang River and Zishui River in Guangxi and the mainstream system of Dongting Lake, has a blocking effect [54].

Therefore, the fish system of Xiangjiang River and Zishui River in Guangxi should be assigned to the South China division under the Southeast Asiatic subregion. Hence, the fish system of the Guangxi water system is classified as the South China division under the Southeast Asiatic subregion of the Oriental region.

### 4.4. Fish Protection

Knowledge of beta diversity patterns can go beyond the systematic conservation planning method, which only considers the location of a protected area to natural, physical, and biological patterns [10,57]. The efficiency of protected areas not only relies on species richness but also on how well the complementarity among sites increases biodiversity conservation [58,59]. In this study, as turnover and replacement components brought larger contributions to beta diversity, additional conservation efforts must target an increase in the number of protected areas, which should be spread across each Sub−basin to maximize the protection of species diversity.

Furthermore, due to the lack of awareness about Guangxi’s freshwater ecosystem (freshwater fish, in particular), the freshwater ecosystem in Guangxi is facing ecological and environmental problems, such as habitat fragmentation, habitat loss, fish migration channel barrier, biological invasion, etc., due to the construction of water conservancy projects and overfishing in recent years [23,40,41]. This destroys the original aquatic ecosystem, reducing the number of freshwater fish and altering the population structure. The water area in the northwest of Guangxi is particularly important, which breeds a large number of cave fishes (HSR, LGR, and LR) endemic to Guangxi—most of them are endangered [23,60]. Currently, only 14 species of cavefish are listed on the Red List of Chinese Species [61], including 9 species of Cyprinidae and 5 species of Nemacheilidae. Since the relevant survey data in this regard are scarce, there are many endangered species in urgent need of protection.

Therefore, attention should be paid to the restoration of the aquatic ecosystem in northwestern Guangxi (HSR, LGR, and LR), and the monitoring and protection of freshwater fish in this area should be strengthened. Obviously, there is a great deal of basic research work to (urgently) be done to establish the situation and evaluation grade of the cave fishes in Guangxi. To determine a protected species, the first step is to complete its biological survey, master its population number, age composition, population structure, growth, fecundity, breeding period, etc., and then design, organize, and implement conservation plans.

## 5. Conclusions

The following general findings were generated from the present study: (1) There are 380 species of freshwater fish in Guangxi, these include 342 native freshwater fish, 128 endemic species, 83 species of cavefish, 49 threatened species, and 18 alien species. The number of freshwater fish species in each Sub−basin generally showed a declining trend from northwest to southeast. (2) Turnover and replacement components brought larger contributions to beta diversity in Guangxi. (3) The freshwater fish system in Guangxi belongs to the South China division in the Southeast Asiatic subregion of the Oriental region. The 12 Sub−basins in Guangxi could be divided into four groups, and the phylogenetic relationships of freshwater fishes in this region generally reflect the connections between water systems and geological history.

## Figures and Tables

**Figure 1 animals-12-01626-f001:**
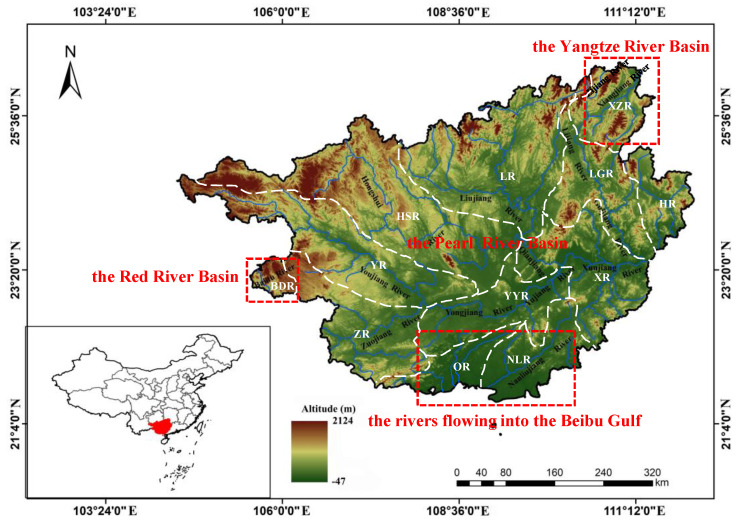
General situation of the freshwater system and Sub−basins in Guangxi. The white dotted lines represent sub−basin boundaries. Sub−basin codes are described in Table 1.

**Figure 2 animals-12-01626-f002:**
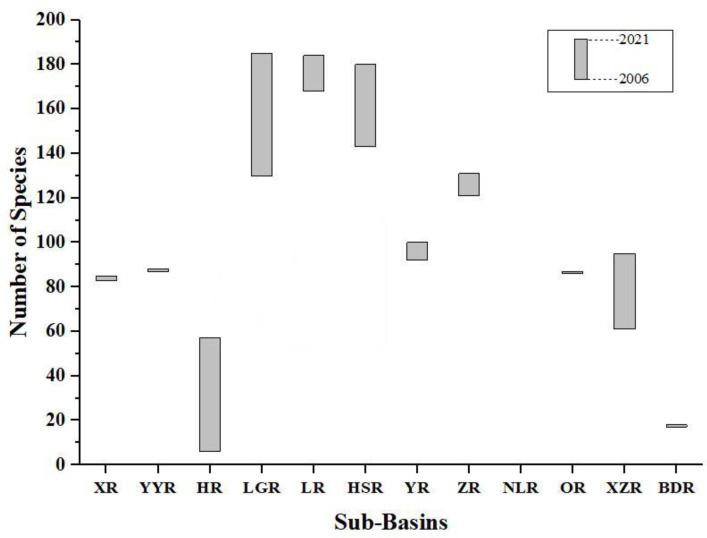
Comparison of fish species numbers in 2006 and 2021. Sub−basin codes are described in Table 1.

**Figure 3 animals-12-01626-f003:**
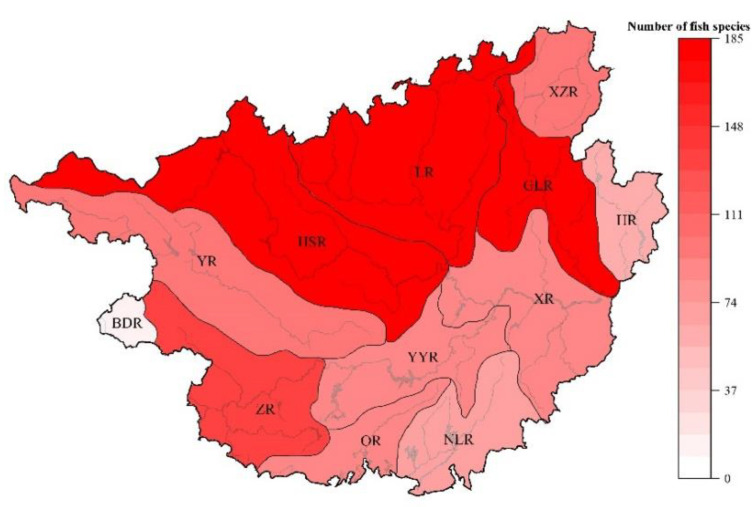
The spatial patterns of fish species numbers in 12 Sub−basins. Sub−basin codes are described in Table 1.

**Figure 4 animals-12-01626-f004:**
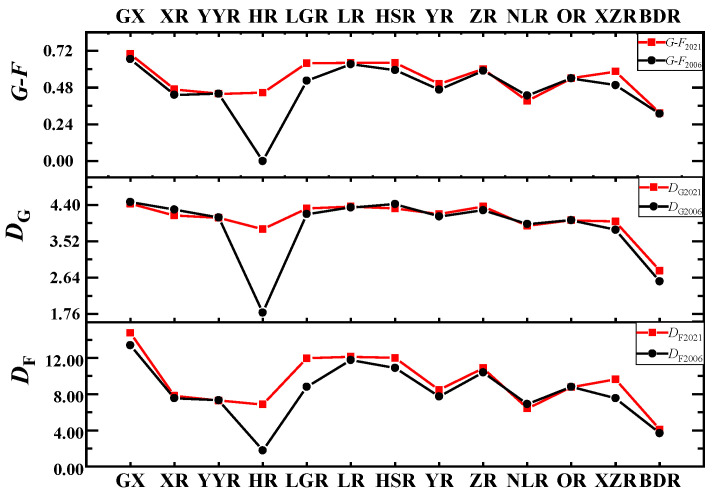
Comparison between 2006 and 2021 at the *G*, *F*, and *G-F* indices of 12 Sub−basins in Guangxi. GX stands for the whole basins of Guangxi; Sub−basin codes are described in Table 1.

**Figure 5 animals-12-01626-f005:**
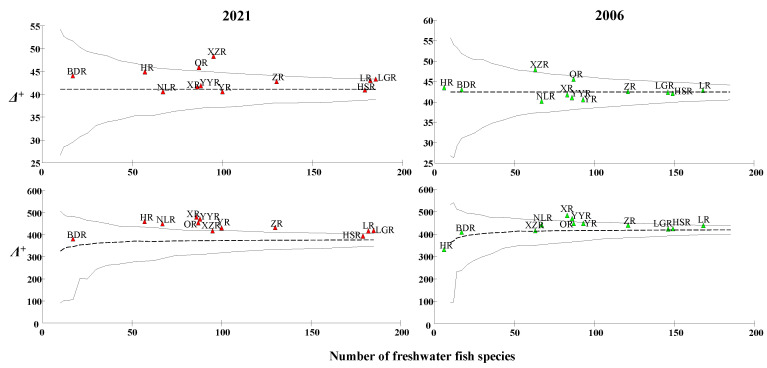
Δ^+^ and *Λ*^+^ of 12 Sub−basins plotted against the number of species on the 95% confidence funnel in 2006 and 2021. (linear-the theoretical value, the two curves-incredible curves).

**Figure 6 animals-12-01626-f006:**
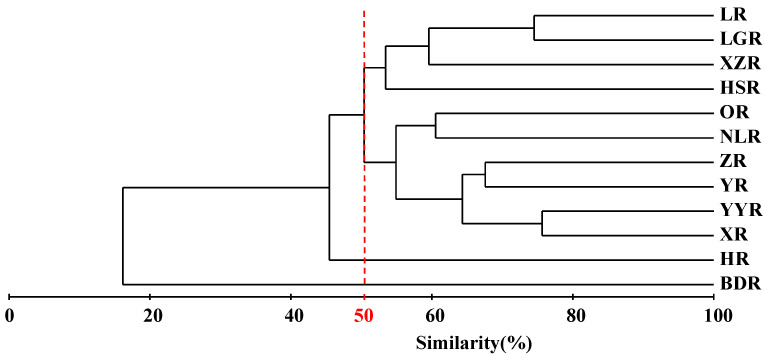
Cluster analysis of 12 Sub−basins of freshwater fish data for Guangxi based on the Jaccard similarity matrix and group average clustering method. Sub−basin codes are described in Table 1.

**Figure 7 animals-12-01626-f007:**
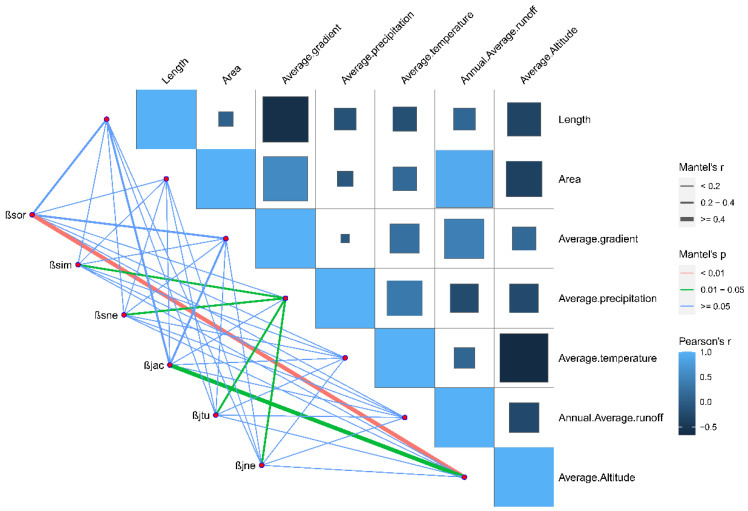
Effects of geographical drivers on the pairwise compositional similarity and the partitioned components obtained from BAS frameworks in 12 Sub−basins of Guangxi.

**Table 1 animals-12-01626-t001:** Hydrological and environmental characteristics of 12 sub−basins in four basins, Guangxi.

River Basin	Sub−Basins Abbreviation	Length (km)	Area (km^2^)	Average Gradient (%)	Average Precipitation (mm)	Average Temperature (°C)	Annual Average Runoff (×108 m^3^)	Average Altitude (m)
Pearl River	XR	291		0.0895	1418	21.6	1230	25
	YYR	426	20,593	0.0199	1320	21.6	410	65
	HR	352	11,536	0.0580	1555	21.7	112	90
	LGR	426	19,288	0.0247	1890	19.8	42	103
	LR	773	51,713	0.0168	1600	19.0	410	99.3
	HSR	659	52,600	0.0380	1100	18.0	696	242
	YR	707	38,612	0.0280	1187	17.1	172	120
	ZR	346	32,068	0.0366	1300	20.7	174	89
Into the Beibu Gulf	NLR	287	9232	0.0377	1458	21.9	77	46
	OR	692	17,322	0.0354	1386	22.0	62	58
Yangtze River	XZR	221	3414	0.0360	1236	17.0	44.9	225
Red River	BDR	62	1758	0.0700	1465	19.2	12	358

Abbreviated instructions: XR: Xunjiang River and Qianjiang River; YYR: Yongjiang River and Yujiang River; HR: Hejiang River; LGR: Lijiang River and Guijiang River; LR: Liujiang River; HSR: Hongshui River; YR: Youjiang River; ZR: Zuojiang River; NLR: Nanliujiang River; OR: Other southern rivers flowing to the sea; XZR: Xiangjiang River and Zijiang River; BDR: Baidu River.

**Table 2 animals-12-01626-t002:** The average taxonomic distinctness (Δ^+^) and variation in taxonomic distinctness (*Λ*^+^) of 12 Sub−basins in Guangxi. Sub−basin codes are described in Table 1.

	Year	XR	YYR	HR	LGR	LR	HSR	YR	ZR	NLR	OR	XZR	BDR
Δ^+^	2021	41.7	42.0	45.0	43.3	43.1	41.0	40.6	42.8	40.6	45.9	48.3	44.1
	2006	41.8	41.1	43.6	42.4	42.9	42.1	40.6	42.6	40.2	45.6	48.0	43.0
*Λ* ^+^	2021	480.9	471.8	460.3	419.4	417.5	394.9	432.0	433.1	449.9	454.8	418.2	381.2
	2006	484.2	470.1	331.1	423.5	439.3	424.4	448.6	440.5	441.2	448.0	416.2	409.1

**Table 3 animals-12-01626-t003:** Fish compositional similarity by BAS frameworks of 12 Sub−basins in Guangxi.

Sub−Basin	β					
	Sørensen Index			Jaccard Index		
	β_sor_	β_sim_	β_sne_	β_jac_	β_jtu_	β_jne_
XR	0.45 ± 0.14	0.29 ± 0.13	0.16 ± 0.14	0.61 ± 0.12	0.43 ± 0.16	0.18 ± 0.15
YYR	0.45 ± 0.14	0.29 ± 0.14	0.16 ± 0.14	0.61 ± 0.13	0.43 ± 0.16	0.18 ± 0.15
HR	0.55 ± 0.11	0.31 ± 0.14	0.24 ± 0.14	0.7 ± 0.08	0.46 ± 0.15	0.25 ± 0.16
LGR	0.47 ± 0.16	0.2 ± 0.09	0.28 ± 0.18	0.63 ± 0.14	0.32 ± 0.12	0.31 ± 0.18
LR	0.47 ± 0.17	0.2 ± 0.09	0.27 ± 0.17	0.62 ± 0.14	0.33 ± 0.12	0.3 ± 0.17
HSR	0.53 ± 0.14	0.29 ± 0.09	0.24 ± 0.16	0.68 ± 0.11	0.44 ± 0.11	0.24 ± 0.15
YR	0.46 ± 0.13	0.3 ± 0.1	0.16 ± 0.12	0.62 ± 0.11	0.46 ± 0.12	0.16 ± 0.11
ZR	0.42 ± 0.15	0.22 ± 0.06	0.2 ± 0.13	0.58 ± 0.13	0.36 ± 0.08	0.22 ± 0.11
NLR	0.51 ± 0.11	0.32 ± 0.11	0.2 ± 0.13	0.67 ± 0.09	0.47 ± 0.13	0.2 ± 0.14
OR	0.5 ± 0.11	0.35 ± 0.1	0.15 ± 0.13	0.66 ± 0.09	0.51 ± 0.11	0.15 ± 0.13
XZR	0.51 ± 0.13	0.37 ± 0.16	0.14 ± 0.12	0.67 ± 0.1	0.52 ± 0.2	0.15 ± 0.15
BDR	0.83 ± 0.03	0.4 ± 0.1	0.42 ± 0.12	0.91 ± 0.02	0.57 ± 0.1	0.34 ± 0.11
Total	0.79 ± 0.05	0.60 ± 0.06	0.19 ± 0.03	0.88 ± 0.05	0.75 ± 0.05	0.13 ± 0.03

Sub−basin codes are described in Table 1.

## Data Availability

Data are available upon request.

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
