# Peer review of "Diversity, Distribution, and Biogeography of Freshwater Fishes in Guangxi, China"

_animals, 2022, doi:10.3390/ani12131626_

Round 1

Reviewer 1 Report

He et al analyzed the diversity of freshwater fishes in Guanxi, providing a clear view of the temporal and spatial patterns of the fishes. The data is detailed and the work is sold. Below are my minor concerns, which may help to improve the MS:

The Sørensen and Jaccard indices provide quite similar information, so it’s redundant to present both results. Consider keeping only one of them.

L15 delete ‘were’

L40 when talking about human activities on the global freshwater fishes, you shouldn’t miss this more recent study:

Su, G., Logez, M., Xu, J., Tao, S., Villéger, S., & Brosse, S. (2021). Human impacts on global freshwater fish biodiversity. Science, 371(6531), 835-838.

L70-73 the expression of the three phrases (aims) should be consistent.

L96 check the table legend

L 116 … looks crowded, add space between items in all equations

L156 add ‘the’ before ‘two’

L165 delete questions marks

L199 add a comma before ‘LGR’

L199 how this test was made if you only have one value per year?

L334-337 this distribution information seems to be not correct if you check it on FishBase

L543 check the author’s surname

Reviewer 2 Report

GENERAL COMMENTS

As its titles refers, the manuscript addresses species composition and diversity of stream fishes in Guangxi, China, by employing diversity and richness metrics between different periods (2006 and 2021), while analyzing spatial distribution between different sub-basin. My first concern of this manuscript is that it is a very descriptive study with local interest, and without any broader scope to other context, other than the Guanxi itself. The authors also compared to different periods (2006 and 2021) and came to the conclusion that the number of species has significantly increased. But what was the cause of such increase? If number of species and diversity was higher from 2006 to 2021, was this related to non-native species invasions and introductions? Was this favoured by the construction of new reservoirs from dams, from where this species typically proliferate? In resume, what could explain such changes in fish community composition? As in many other parts of the world, I suspect that this Is due to river regulation and dam building, that facilitate invasion by non-native species, while decreasing the number of native and endangered ones. All this aspects should be better explored in the revised version. Hence, must be better described and point out the main human pressures in the basin.  The authors refer that “The results will provide the information needed for freshwater fish resource protection in Guangxi”. In what way? What specific measures could be applied here (and in other regions) to contribute to fish protection and conservation? Seems to me this should consider dam removal or retrofitting, the building of fish passes, among others. Test statistics are poorly described, with no indication of the tests or statistics. Some figures and tables are redundant to each other. Further, there are several parts of the manuscript that are too long, descriptive and boring to read (for details see specific comments below), and need to be shorten (discussion is far too long and should be shortened atr least on 1/3, keeping in trying to respond to the study goals outlined on the final of the Introduction. Caption of figure and tables, are mostly vague and lack details. Besides all this aspects, my major concern of the paper retains, i.e. in that this a very descriptive study of local interest. Could the improve the scope of it? This must be addressed if the paper wants to be consider for publication. Make yourself the following question: how can the findings of my paper be useful to other readers, of other regions, other than Guangxi?

SPECIFIC COMMENTS

L65 – “new species and records have also been reported”. Non-native/invasive species?

L68 – avoid the use of “etc.”

L71-72 – “(2) what contributes most to beta diversity: replacement or nestedness?” Why beta-diversity? Why this metric is particularly important for the study? This should be outlined before, perhaps on L50 when you refer to it.

L73 – “process of freshwater fish in Guangxi.”. This is too vague. Processes that drive fish community composition and/or structure? Other?

Figure 1 – Please include the north indication (typically an arrow).

L88 – Could you provide a reference?

L90 – Reference?

L90-94 – But where is this information? Table? Figure?

L96 – please check table caption.

L103 – Situation is too vague. You are showing the river network and elevation range of the basin.

L107 – What are the zones? Figure 1 does not show them.

L108-109 – Could you provide references for these indicators?

Table 1 – Not sure this table is necessary, as you do not show such detail of subbasins on Figure 1.

L128-130 – Please provide a reference.

L134 – punctuation – use “:” instead of “.”

L153 – Add reference of the PRIMER package.

L199 – What statistical test was done here? Do not recall to see this on M&M

Figure 2, y-axis: replace “Species” by “Number of Species”. Do not really understand what this figure wants to show: The bars represent the range of number of species from 2006 to 2021? So for example, for HR, in 2006 there were c. 8-9 species and on 2021, there were more than 50? I believe this figure could be removed from the manuscript, and such numbers could be included on Table 1.

L201 – The caption is very poor and vague (as others in the manuscript). What do the letter on the X-axis represent? Where such information can be found on the manuscript? What do the asterisks represent? The x-axis is named “Rivers”, but on Table 1, such codes are named as sub-basins. Check and correct!

L205-208- please be specific. Provide numbers, name of test, test statistics between parenthesis.

Figure 3, L210 – What spatial patterns? What do you mean by this? I only see a map of red colors without any caption, units, variables, etc. What do the acronyms represent?

L211 – The top 8 of what?

L229 – “Among them”? Who?. Never start a paragraph with such words.

L240 – You already said this in M&M

L281 – The figures should be shown on Figure 6.

L292 – P value and test statistics?

L293-298 – Where are (figure? Table?) these correlations shown? The authors present no values at all, neither test statistics.

L312 – Sub-titles of the Discussion should be re-written according to the study objectives.

L323-324 and throughout the manuscript – you must never discuss results, without first bering described in the appropriate section (Results). This result “There are 13 newly recorded exotic fish species such as Prochilodus lineatus, Tinca tinca, Pterygoplichthys pardalis, etc.”, was not previously described. Describe it now on the Results or remove the sentence from the manuscript. This applies also to the rest of the Discussion.

L324 – Replace “invasion site” by “recipient site”.

L327 – Most importantly, it would have been highly useful, if the authors have analysed how non-native species increased from 2006 and 2021. If the number of species increased between both years, as the authors show, does this refer to new introductions (i.e of non-natives (invasive) species). This should be the focus of the results.

L328-353 – This section is too descriptive, boring to read, and it seems it deviates from the main results.

L382-384 – Changes in what sense? If number of species and diversity was higher from 2006 to 2021, was this related to non-native species invasions and introductions? Was this favoured by the construction of new reservoirs from dams, from where this species typically proliferate? In resume, what could explain such changes in fish community compostion?

L394 – What do you mean with the “single habitat”? Which is it? Did not see this on M&M.

L401-402 – Avoid speculation in provide references to support your statements.

L413-422 – This is seems a too much simplistic explanation. Please keep to the discussion of the main results, trying to answer the 3 goals that you addressed at the end of the Introduction.

L434-469 – Again, these long explanations of the history geology of the study area, deviates from the study objectives, which should focus to  try to answers the goals of the study.

479-484 - Why was the original aquatic ecosystem destroyed? What are the main pressures that the study area faces? Dams construction (I suppose)? River regulation? Organic pollution? Other? This should have been previously assessed on the Study Area.

L489 – To which sub-basins does this corresponds?

L500- Discussion section did not answer the goal previously outlined on the objectives: (2) what contributes most to beta diversity: replacement or nestedness?. Please clarify this here and also on the previous discussion section.

Round 2

Reviewer 2 Report

I am satisfied the authors comments to previouc concerns, as well as with the revised version, which significantly improved over the original submission.

I believe the manuscript can now be accepted as it stands.